# Exoproteomics for Better Understanding *Pseudomonas aeruginosa* Virulence

**DOI:** 10.3390/toxins12090571

**Published:** 2020-09-04

**Authors:** Salomé Sauvage, Julie Hardouin

**Affiliations:** 1Polymers, Biopolymers, Surface Laboratory, UMR 6270 CNRS, University of Rouen, CEDEX, F-76821 Mont-Saint-Aignan, France; salome.sauvage@univ-rouen.fr; 2PISSARO Proteomics Facility, IRIB, F-76820 Mont-Saint-Aignan, France

**Keywords:** exoproteomics, *Pseudomonas aeruginosa*, virulence factors, post-translational modifications, secreted proteins

## Abstract

*Pseudomonas aeruginosa* is the most common human opportunistic pathogen associated with nosocomial diseases. In 2017, the World Health Organization has classified *P. aeruginosa* as a critical agent threatening human health, and for which the development of new treatments is urgently necessary. One interesting avenue is to target virulence factors to understand *P. aeruginosa* pathogenicity. Thus, characterising exoproteins of *P. aeruginosa* is a hot research topic and proteomics is a powerful approach that provides important information to gain insights on bacterial virulence. The aim of this review is to focus on the contribution of proteomics to the studies of *P. aeruginosa* exoproteins, highlighting its relevance in the discovery of virulence factors, post-translational modifications on exoproteins and host-pathogen relationships.

## 1. Introduction

*Pseudomonas aeruginosa* is a common environmental Gram-negative bacterium, which is part of the *Pseudomonadaceae*. This bacterium is aerobic, motile, non-sporing, and ubiquitous because of its ability to survive in large environment range [1]. This universal distribution suggests a remarkable degree of genetic and physiological plasticity to environmental changes [2].

*P. aeruginosa* is also the most common human opportunistic pathogen associated with nosocomial diseases [2]. In 2017, the World Health Organization has classified *P. aeruginosa* as a critical agent threatening human health and for which the development of new treatments is urgently necessary [3]. Indeed, *P. aeruginosa* is able to avoid the innate host immunity and is highly resistant to a wide range of antimicrobial agents [4], making its infections very difficult to treat [2]. Discovering of new antibiotic drugs can be a solution. Nevertheless, new bacterial resistance can appear [5]. A strategy more and more considered is to reduce bacterial virulence, i.e., not kill the bacteria [6]. With this anti-virulence strategy, the selection pressure that could be exerted does not engage bacterial survival, suggesting that this type of strategy would limit the selection of resistant strains. In this context, the study of the virulence factors is an interesting avenue to understand *P. aeruginosa* pathogenicity.

*P. aeruginosa* possesses a large arsenal of virulence systems contributing to the success of its infection and colonization [7]. Among them, we can cite quorum sensing (QS) and type 2, 3, and 6 secretion systems (T2SS, T3SS, and T6SS respectively) with their secreted proteins and toxins. QS is an intercellular bacterial communication system dependent on cell density, which allows individual cells to act as a community [4]. This system would control around 10% of the *P. aeruginosa* genome, including survival genes, genes coding for virulence factors and biofilm formation [7,8,9]. T2SS, T3SS, and T6SS are major secretion systems in *P. aeruginosa* (Figure 1). Their secreted products act as virulence factors: they support bacteria to survive in host by avoiding the immune host system and allowing them to provoke infection [4]. T2SS secretes several proteases in the extracellular medium, including elastases LasA and LasB, protease IV, phospholipase C, and exotoxin A [7]. LasB is a metalloprotease able to destroy or inactivate biological tissues in host organisms [10]. LasA is involved in proteolysis and degradation of the host protein elastin and enhances the elastolytic activity of LasB and other proteases. Protease IV and phospholipase C participate in *P. aeruginosa* keratitis and *P. aeruginosa* acute lung injury and inflammation, respectively [7]. Exotoxin A inhibits protein synthesis by its ADP-ribosyl transferase activity leading to cell death and depresses host response to infection [7]. All of these secreted proteins can be exported through the cytoplasmic membrane via Sec or Tat machinery [11]. T3SS consists in an injectisome, synthetized, and assembled on the bacterial surface, when the bacterium is next to host cells [12]. It injects toxins directly in the cytoplasm of target cells and four exotoxins have been identified in *P. aeruginosa* strains: ExoS, ExoT, ExoU, and ExoY. ExoS and ExoT target and inactivate different substrates including GTPases and adaptor proteins, which trigger actin cytoskeleton dismantlement and then cell apoptosis [12]. ExoU permeabilizes the plasma membrane and leads to necrotic cell death by its phospholipase activity [12]. The toxicity of ExoY, an adenylate cyclase, remains unknown [12]. T6SS is composed by core-conserved genes (TssA-C, TssE-G, TssJ-M, Hcp, VgrG, and ClpV) which form its key structures [13]. It is able to inject effector proteins inside eukaryotic host cells, prokaryotic competitors and the environment, providing fitness and survival advantages to *P. aeruginosa* [14,15,16]. Indeed, T6SS has a large range of cellular targets and uses an arsenal of toxins and effectors to subvert or kill prey cells. Thus, three T6SS clusters named H1-T6SS, H2-T6SS, and H3-T6SS are encoded by *P. aeruginosa* [14]. Among them, H1-T6SS transports at least seven antibacterial toxins (Tse1-Tse7) [14], which target membrane, cell wall, nucleic acids and have other biological functions [14]. H3-T6SS is involved in iron acquisition [15,16] and antibacterial activity [17]. Moreover, T6SS is usually not expressed suggesting that environmental factors or host colonization trigger T6SS assembly [4].

Several cytoplasmic proteins (chaperonin GroEL, chaperonin DnaK, glyceraldehude-3-phosphate dehydrogenase, superoxide dismutase A (SodA), etc.) have also been described in the extracellular compartment. They are secreted by non-classical protein secretion pathways, for which secretory components are not yet well known [18]. Interestingly, the functions of these proteins inside and outside the cells are not the same, and some of them are involved in cell defence or virulence [19]. Exotoxins have, thus, different modes of action giving bacteria a wide range of virulence mechanisms [20].

Excessive use of antibiotics during treatment accelerates the development of *P. aeruginosa* multi-resistant strains, leading to the ineffectiveness of antibiotic therapy [2,8]. Consequently, identifying new therapeutic targets is becoming an emergency to develop more effective treatments. Secreted proteins are implicated in essential functions like biofilm formation, iron uptake, motility or virulence [21]. Recent studies have reported that several new non-antibiotic therapeutic approaches, such as QS inhibition, bacterial lectins inhibition, phagotherapy, or vaccine strategy appear efficient to eliminate *P. aeruginosa* antibiotic-resistant strains [2]. Indeed, disruption of QS mostly inhibits the production of secreted proteins, resulting in the inability for the bacteria to metabolise extracellular nutrients [22]. Post-translational modifications (PTMs) on a protein may allow bacteria to answer in a short term to internal and external signals [23,24]. Their control would also prevent microorganisms from reacting to these signals and reduce their fitness. Therefore, focusing on virulence factors can be an interesting strategy to fight against *P. aeruginosa* [6]. Molecules inhibiting virulence factors activities (such as proteases) can be an additional treatment for *P. aeruginosa*-infected patients in order to decrease host inflammation [25]. Therefore, the characterisation of exotoxins is an important research topic.

Proteomics is now a well-established strategy to characterise proteins in a cell, a tissue or an organism (eukaryotes, prokaryotes, plants, etc.). In the last decades, numerous bacterial genomes have been sequenced, allowing large-scale proteomic studies in bacteria, including pathogenic bacteria such as *P. aeruginosa*. Two aspects should be considered for proteomic analysis: the tremendous technological developments and the useful information on the identity of proteins present in different conditions and the highlighting some molecular mechanisms established by the bacterium under these conditions. Concerning the technological point of view, proteomics leads on three important steps: sample preparation, analysis by nano liquid chromatography-tandem mass spectrometry (nanoLC-MS/MS) and bioinformatic interpretation. The advances in instruments, bioinformatic tools, and sample preparation kits have allowed researchers to gain sensitivity, resolution, and speed in data acquisition. This review will not provide a state-of-the-art review of these technical advances. Beside these improvements, proteomics is a powerful tool allowing protein identification and quantification, PTMs characterisation, intact protein detection, or non-covalent complexes study. By highlighting biochemical pathways of interest, the proteomic data pave the way to other more specific biological experiments. With bacterial genome sequencing, proteomics has emerged as a major approach.

Many proteomic analyses have been performed to identify and quantify intracellular proteins in *P. aeruginosa* in different conditions, which have contributed broadly to the understanding of bacterial physiology and virulence [26,27,28,29,30,31]. However, these studies miss an important subset of proteins that are secreted into the extracellular milieu [32]. They are useful for a better understanding of *P. aeruginosa* virulence. Moreover, they constitute potential biomarkers and therapeutic targets for the development of new antibacterial strategies to fight against pathogens such as *P. aeruginosa* [33].

Proteomics is a strategy of choice to perform in high-throughput virulence factor identifications in different conditions and to highlight their effects on host cells [27]. The purpose of this review is to provide an overview of the contribution of proteomics in the knowledge of *P. aeruginosa* secreted proteins. It focuses on the exoproteomes of *P. aeruginosa*, i.e., the proteomic analysis of proteins that are secreted outside the cell (named exoproteins) [34]. In the first part, we will describe the sample preparation steps to be taken in consideration in order to successfully analyse the exoproteins. The second and third sections will focus on the relevance of proteomics in the characterisation of both *P. aeruginosa* exoproteins and their PTMs. Finally, the last section will consider the contribution of proteomics to a better understanding of host-pathogen relationships.

## 2. Keys to Successful Proteomic Analyses of Exoproteins

As described above, *P. aeruginosa* has developed a variety of secretion systems. The exported proteins as well as the components of the secretion apparatus can be targets of interest for drug development [21], and their identification is thus a critical initial task to understand bacterial virulence. However, the characterisation of exoproteins is a real challenge.

Different approaches have been developed and are available: biochemical, bioinformatic, genomic, imaging tools, or proteomics [35]. Bioinformatic analysis is a robust and fast approach, but it is limited to secretion systems for which prior information is known [16]. The use of machine learning classification algorithms can thus identify new effectors of secretion systems [36]. To know the real genetic dynamics following to a stimulus, analysis of the gene products (e.g., proteins) is necessary. Currently, proteomics is a powerful strategy to identify and quantify proteins in high throughput, including effectors and secretion apparatus components. For example, Fritsch et al. successfully identified four new proteins secreted by T6SS in *Serratia marcescens* by proteomics [37]. Proteomics presents thus the great advantage to identify effector proteins either without knowing the secretion system from which these effectors originate, or from a specific secretion system without any prior knowledge.

Proteomics can give answers to different questions (identification, quantification, PTMs, macromolecular complexes, etc.), but these questions have to be clearly asked to define the best experimental design from bacterial culture to protein identification (Figure 2). Proteomics presents the great advantage to identify effector proteins either without knowing the secretion system from which these effectors originate, or from a specific secretion system without prior any knowledge. One downside in this research area is the dynamic range. Effectors have to be secreted and in a significant amount to be detected [16]. To improve protein secretion and thus their identifications, different experimental conditions can be set up, especially for secretion systems triggered by a cell contact. In *P. aeruginosa*, T3SS is induced in vitro by calcium depletion. This depletion mimics the cell medium in which calcium is quenched and this signal is sufficient to induce T3SS [12,38]. T6SS is activated by close contact with target cells for interbacterial competition or eukaryotic host pathogenesis [39]. This contact is difficult to make and exoproteomes of T6SS-deficient mutants and wild type (WT) strains are generally compared in order to identify effectors [16]. Another important point is to know when virulence factors are the most expressed in order to improve their characterisations. In *P. aeruginosa*, several are expressed during the stationary phase [40,41]. Specific conditions are needed for the secretion of some virulence factors. For example, alkaline phosphatase is expressed in an environment limited in phosphate [42]. Cell density plays an important role as well for protein secretion. Arevalo-Ferro et al. studied QS regulated proteins in *P. aeruginosa* at the late exponential growth phase where most of these proteins are induced [43]. Temperature also influences secretion of virulence factors as shown in different studies [44,45,46]. According to the culture medium, proteins secreted in the supernatant can change as exemplified with the extracellular protein profiles of *P. aeruginosa* PAO1, a laboratory associated burn wound isolate, grown in Luria-Bertani (LB) broth, in M9 medium, or ABt minimal medium supplemented with sodium citrate and casamino acids [43,47,48]. LasB, one of the most abundant virulence factors, was detected with a lower abundance in the two latter media.

After the focus on cell culture, the success of virulence factor identification relies then on careful proteomic sample handling. It requires specific sample preparation in order to eliminate bacteria and the extracellular proteins retrieval from the bacterial supernatant. Proteins are present in a high volume and their precipitation with trichloroacetic acid (TCA), methanol or acetone is a necessary step [48]. One has to remember that, despite the care taken in sample preparation, some contamination due to bacterial cell lysis can occur [40]. However, observation of cytoplasmic proteins found in the extracellular medium does not always mean that cell lysis or sample contamination occurred (see below). Generally, soluble extracellular proteins are well recovered in contrast to less soluble proteins that remain difficult to be characterised. Recently, Lampaki et al. described a sample preparation protocol to improve the proteomic analysis of T3SS secretion proteins in PAO1 [49]. They compared three sample preparations: TCA, a solid phase enhanced sample preparation method (SP3) and TCA/SP3. The last approach gave better results, in term of peptide separation and number of protein identifications, to study exoproteomes in which proteins are in low concentration.

Next, the bottom-up strategy (or shotgun) is mainly used for the characterisation of exoproteins in bacteria. Recent reviews have been published describing the different available proteomic approaches for protein identification [50,51,52]. Briefly, proteins are digested with a specific endoprotease (generally trypsin) to yield peptides that will be analysed by high-sensitive and high-resolution nanoLC-MS/MS. After databank searches with bioinformatic tools, both peptides and proteins sequences are identified. Before tryptic digestion, proteins can be separated by one-dimensional gel electrophoresis (SDS-PAGE), two-dimensional gel electrophoresis (2-DE), or liquid chromatography for improving protein identifications.

Besides, other MS-based strategies can also be used in order to provide additional pieces of information to better characterise a biological system. T6SS effectors have been described as being bound to different proteins (Hcp, VgrG, chaperone, etc.) [16]. Affinity purification followed by MS can be performed for effectors identifications [53,54]. Protein-protein interaction studies by native MS can also give insight on new effectors [55], and determine complex stoichiometry, such as for PcrV/PcrG complex in *P. aeruginosa* [56].

Bacterial culture conditions therefore greatly influence the secretion of *P. aeruginosa* virulence factors. It is therefore important to monitor these conditions in order to optimise their identification by proteomic analysis. Many works have been published using these approaches to identify successfully exoproteins.

## 3. Proteomics for Exoproteins Characterisation

### 3.1. Gel-Based Proteomics

Gel-based proteomics, i.e., proteins separation by 2-DE or SDS-PAGE before MS analysis, was initially used for the supernatant characterisation in *P. aeruginosa* (Figure 2). The first study on secreted proteins in *P. aeruginosa* was performed in 2002 by Nouwens et al. [40]. They compared for the first time, by 2-DE and MS, the extracellular proteins of two *P. aeruginosa* strains: the invasive PAO1 and the cytotoxic 6206 strains. The two-dimensional (2D) maps comparison clearly highlighted different protein patterns for the two strains. The 2D extracellular protein profiles of a cystic fibrosis (CF) lung-adapted *P. aeruginosa* strain C and a burn-wound isolate PAO, at the stationary phase, were also not similar [57]. In the same way, Wehmhöner et al. [41] showed that the 2D gels of five *P. aeruginosa* strains isolated from cystic fibrosis (CF) patients were highly different. These different patterns highly complicate both protein assignments and comparisons between the strains by 2-DE. This complexity increases with growing culture time since 2D gels became more and more complex from mid-exponential phase to stationary phase cultures [41]. Collectively, these studies clearly show, by gel comparisons of extracellular protein expression profiles of various *P. aeruginosa* strains (CF patients, burn-wound isolate, laboratory strain), the great ability of this pathogen to adapt to the different niches.

Despite assignment difficulties for a same protein, spot intensities between different conditions were compared in several studies, providing information on protein quantification. Among the common spots between gels, the CF lung-adapted *P. aeruginosa* strain C presented higher amount of the protease IV, involved in corneal infection, whereas spot intensities for LasB were the same for the two strains [57]. The various morphotypes of an isolate from a CF patient were studied by 2-DE [41]. Once again, different protein spots were visualised on gels, suggesting a specific protein regulation in each strain. In this study, they compared the two morphotypes (hyperpiliated SCV 20265 strain and the clonal revertant) to the WT and the abundance of 52 protein spots were then measured. In the exoproteome of the hyperpiliated SCV 20265 isolated from the CF patient, both virulence factors secreted by T3SS (ExoS and ExoT) and the components of T3SS were observed upregulated. Furthermore, T1SS virulence factors HasAP, AprA, and AprX were also identified. These studies revealed one more time that exoproteomes are thus specific to morphotypes and strains.

Two QS systems, *las* and *rhl*, were proposed as a global mechanism to control the virulence factors expression and biofilm development in *P. aeruginosa*. To determine the influence of *las* and *rhl* on the exoproteome profile, two studies were performed in 2003 [43,48]. They together highlighted the *las* and *rhl* influence on the protein pattern of *P. aeruginosa* PAO1 by comparing the WT strain and Δ*las* and/or Δ*rhl* mutants by 2-DE and MALDI/MS. The 2D gels displayed multiple spots corresponding to different secreted proteins between these two studies, carried out in different culture media. Together, the authors concluded that exoproteins abundance was highly reduced in the mutants. The absence of *las* or *rhl* gene interferes therefore virulence factors secretion, like LasB.

The nucleoside diphosphate kinase (NDK) is a kinase that is involved in the regulation of cellular development [58]. It was demonstrated that NDK is also secreted by T1SS in the extracellular medium [59,60]. This ubiquitous enzyme is involved in host inflammatory responses, with the help of flagellin [61]. In 2014, Needl et al. studied T3SS-secretion proteins of the strain PAK-J [62]. By comparing the exoproteome of the WT and a mutant deleted of the three T3S effectors (*exoS*, *exoT* and *exoY*), they showed by SDS-PAGE that the exoproteins patterns were different. After MS identification, they concluded that NDK was a new T3SS effector in *P. aeruginosa*. NDK can thus be secreted by two secretion systems: in the extracellular medium by T1SS and in the host cells in a T3SS-dependant manner. NDK was also identified in the extracellular milieu of the strains PAO1 [40] and PA14, a virulent wound isolate [63].

Moreover, 2-DE and SDS-PAGE have the great benefit to allow the visualisation of proteins and the determination of protein abundance by gel comparisons using software packages. Another advantage for 2-DE is the ability to distinguish spots with different isoelectric points (pI) and/or molecular weights (MW) for a same protein, suggesting the presence of PTMs [50]. Due to the presence of proteases in bacterial supernatants, a protein may be present at different MWs. Furthermore, 2-DE is a unique tool that provides valuable information on the status of proteins in the sample [48]. This technique also has some drawbacks. The presence of the different protein forms makes the gel comparisons difficult [40]. In gel-digestion can be ineffective and tryptic peptides can remain trapped inside the gel, leading to the MS signals absence. Proteins with low- or high-mass, low or high pI can be lost during migration, and the presence of contaminants (keratins, albumin for example) can hide protein spots [40]. Another disadvantage is that these are laborious experiments to perform and therefore repeatability may be impaired.

Despite these drawbacks, 2-DE remains one of the most widely used approaches for investigating proteome of a biological system, especially since the introduction of the Differential Gel Electrophoresis (DIGE) method [64]. This technique makes it possible to compare two samples in a gel with a small quantity of proteins. It improves both the significant labour and the repeatability of gel migration. In an elegant work, Ball et al. chose this approach to determine new Tat-dependant exoproteins in *P. aeruginosa* PAO1 [65]. For this, they compared the WT and the Δ*tat* mutant exoproteomes (strains grown in phosphate starvation inducing virulence factors synthesis) using DIGE followed by protein identifications by MALDI/MS. They identified proteins known to be secreted by T1SS, T2SS, T3SS T5SS, and T6SS and others without known secretion pathways, including hypothetical proteins. Focusing on three hypothetical proteins (PA2377, PA2699, and PA4140) and PA3910 (alkaline phosphatase), the proteomic results were corroborated by other approaches (mutants, immunoblotting, qRT-PCR, in silico computational methods). They concluded that PA2377 is a new T2SS/Xcp dependent exoprotein, and together PA2699, PA3910, and PA4140 are new Tat dependent exoproteins. These proteins are therefore new virulence factors of interest for attenuating virulence in this pathogen. In this work, it was shown that the GlpQ protein is not a Tat-dependent exoprotein, unlike previous results [11]. This study demonstrates once again the power of proteomic analysis to identify new virulence factors unknown as such until now.

The exoproteomes of *P.* aeruginosa laboratory strain PAO1 and *P. aeruginosa* CF isolate AES-1R (Australian epidemic strain, transmissible acute CF clinical isolate) were studied by 2-DE. In this study, Scott et al. used a CF lung-like artificial sputum medium (ASMDM), to be as close as possible to in vivo conditions [47]. However, the presence of serum proteins and mucins was a problem for 2-DE. They did however highlight some differences in protein presence and abundance between the two strains grown in the LB or M9 media. To gain information, the authors used a gel-free approach with 2D-LC-MS/MS. Some virulence factors have been identified in only AES-1R isolate (ChiC and the hypothetical protein PA1592), in PAO1 (AprA, CdrA, and phospholipase C) or in the two strains (LasB, PrpL and aminopeptidase). Quantitative proteomics using Selected Reaction Monitoring (SRM) was undertaken for ten proteins. Five proteins (LasB, CbpD, PasP, PA4495, and PA1342) were more abundant in the AES-1R cultured in ASMDM in contrast to PAO1, suggesting that abundant virulence factors are important for lung infection.

Moreover, 2-DE and LC-MS/MS can be applied in the same study to obtain complementary information. Gel-based approach shows protein variants and gel-free approach gives insight on overall protein abundance without considering each protein form for a same protein. Gel-free strategy is another main approach used in proteomics.

### 3.2. Gel-Free Proteomics

Today, gel-free approaches are widely used, especially since the dazzling advances in MS allowing both protein identification and quantification in a complex sample without the need of fractionation steps (Figure 2). This direct strategy presents many advantages, such as reduced protein and peptide losses, or reduced sample preparation time. However, it is not possible to estimate the minimal number of proteoforms as for 2D map and to see cleaved proteins.

Using gel-free proteomics, Hood et al. focused on the identification of T6SS virulence factors in *P. aeruginosa* PAO1 [66]. However, the characterisation of T6SS effectors is a real challenge since cell contact is needed to activate T6SS secretion. The authors compared, thus, the mutants’ Δ*pppA*, to induce T6SS secretion, and Δ*clpV*, to inhibit T6SS assembly. Three hypothetical proteins (PA1844, PA2702 and PA3484) were revealed as new T6SS substrates, called Tse1-3. These MS identifications led to perform complementary experiments (fluorescence microscopy, infection assays, competition assays). These results corroborated the proteomic data proving the T6SS secretion of these hypothetical proteins. The authors demonstrated that *P. aeruginosa* uses Tse2 to compete with bacterial cells but not eukaryotic cells.

Quantitative proteomics gives additional insights of interest on protein abundance in various conditions. To determine when virulence effectors are the most expressed, quantitative proteomics can be performed. The kinetics of three T3SS virulence factors export (ExoS, ExoT, and ExoY) were studied by label-free quantification in *P. aeruginosa* PAO1. A higher secretion level was determined at two hours after induction [49]. Recently, Lin et al. used SRM approach and stable-isotope-labelled peptides to determine the abundance of the components of the T6SS in three bacteria *Vibrio cholerae*, *Acinetobacter baylyi,* and *P. aeruginosa* (strain PAO1) [67]. For the latter, the results showed that the protein abundances matched with the T6SS stoichiometry since ClpV was lower in abundance than TssB and TssC.

Bergamini et al. studied the laboratory strain PAO1, an early *P. aeruginosa* clinical isolate and a late *P. aeruginosa* clinical isolates by Multidimensional Protein Identification Technology (MudPIT) approach to characterise exoproteins involved in pro-inflammatory activity [68]. This large-scale proteomic analysis identified several metalloproteases in the supernatant. It shows the *P. aeruginosa* ability to induce pro-inflammation in bronchial epithelial cells.

The in vivo virulence on *Galleria mellonella* model of seven paired genetically indistinguishable clinical bloodstream and peripheral isolates of *P. aeruginosa* showed that the bloodstream isolates are more virulent than the peripheral isolates [69]. To characterise virulence factors involved in this virulence, the exoproteomes of these two isolates were compared after nanoLC-MS/MS analysis. The abundance of 93 proteins were differentially measured with adhesion factor LecA and alternative sigma factor RpoN (regulating different virulence factors) in higher abundance in the bloodstream isolates. A total of 53 proteins with unknown functions were also observed up- or down-regulated. The infection microenvironment can thus influence the expression of virulence factors in *P. aeruginosa*.

These examples have clearly shown that proteomic studies of bacterial supernatants have the ability to reveal proteins as potential virulence factors, which would not have been suspected without these analyses, as hypothetical proteins. These results often pave the way to other microbiological or biochemical experiments to confirm the involvement of these proteins in virulence. However, MS-based approaches can also be used after these experiments to identify or to follow proteins of interest. For example, Casilag et al. showed that *P. aeruginosa* has more than one protease (AprA) involved in the degradation of flagellin to avoid immune recognition [70]. Using native PAGE for studying flagellin degradation of both PA14 WT and Δ*aprA* exoproteins, a protein band was revealed and identified as the protease LasB by SDS-PAGE and MS. Thus, LasB is involved in a cooperative relationship with AprA in the anti-flagellin activity.

High-throughput comparison of bacterial samples has become a routine procedure in proteomics allowing the identification of both new effectors, constituents of secretion systems, and variations in the expression of exoproteins. It is, therefore, a valuable support for understanding the virulence mechanisms involved in very specific conditions.

### 3.3. Exoproteins Subcellular Localisation

Subcellular localisation of identified exoproteins is generally searched to ensure their secretion with different prediction tools [32]. Amongst all these gel-based and gel-free studies, proteins identified in extracellular media have mainly cytoplasmic predicted localisation. However, protein localisation prediction tools are not always reliable, and more and more cytoplasmic proteins are found in the extracellular compartment. The prediction tools rely on secretion motifs as peptide signals to assess the secretion of a protein. However, many proteins are not yet predicted as secreted proteins because no Tat-peptide or Sec-peptide signals, for example, is determined in the protein sequence while they have been identified in the extracellular medium [65]. Wrong annotations of the genome can result in incorrect predictions for protein localisation and it is estimated that 15% of the genes are not properly annotated [71]. Furthermore, it is now known that non-classical protein pathway exists in bacteria, but the secretion process is still unknown [18]. Several proteins have been called moonlighting proteins since they show different functions [72,73]. Some of them have been described as secreted, such as DnaK, GroEL, or SodA [74]. Therefore, the high throughput screening of extracellular compartments can be informative for new virulence factors discovery.

### 3.4. Hypothetical Proteins as New Virulence Factors

The *P. aeruginosa* genome encodes about 40% of hypothetical proteins. Proteomics is a major asset for verifying the existence of these proteins, with unknown function. A substantial percentage of hypothetical proteins are usually identified in exoproteomes (around 20–30% of identified proteins). These proteins are thus expressed by *P. aeruginosa* and their localisation can be determined.

Recently, as described above, the three hypothetical proteins PA2377, PA2699, and PA4140 were highlighted as secreted proteins via Tat secretion pathway by proteomic analysis [65], while no Tat-signal peptide can be determined in their sequences. Additional experiments will be necessary to determine their involvement in *P. aeruginosa* virulence.

Interestingly, a same hypothetical protein can be identified in several *P. aeruginosa* extracellular protein studies conducted in different culture media and with different proteomic approaches. This can suggest that they are real secreted proteins, for which we have to determine their role in bacterial virulence. An example is the hypothetical protein PA0423. PA0423 was identified in three exoproteomes of both *P. aeruginosa* PAO1 [40,43,47] and PA14 [63]. During the growth-inhibitory phase, Shinagawa et al. also characterised this protein as a bacteriolysis-associated virulence factor by using different separation steps (gel filtration and strong anion exchange) followed by MALDI/MS and A549 cell-proliferation inhibition assays [75]. PA0423 is annotated PasP, a serine protease that has been reported as a virulence factor in different studies on keratitis and human respiratory tract infections [76]. Recently, this protein was described acetylated or succinylated on lysine 42 in PA14 strain [23]. This PTM could play a role in the function of this protein.

## 4. Proteomics for the Characterisation of PTMs Involved in Virulence

The addition and the removal of small chemical groups on specific amino acids are defined as PTMs [77]. The most studied modifications in bacteria are phosphorylation and acetylation. PTMs are a real advantage for bacteria because they allow them to adapt quickly to environmental changes and thus to survive in different conditions. Currently, it is well recognised that PTMs play a crucial role in multiple bacterial pathways [24] as cell division [78], metabolism [79], bacterial resistance [80], biofilm formation [81], persistence [82] or virulence [83,84].

Previously, PTM characterisations were performed at a low-throughput level. Faced to their importance, high-throughput approaches have been pivotal in rapidly identifying modified proteins. Nowadays, proteomics is the tool of choice to identify, quickly and reliably, diverse modifications (Figure 2). In the past two decades, PTMs have been successfully characterised in intracellular medium by different proteomic strategies previously reviewed [50,85]. In *P. aeruginosa*, the studies of phosphorylation [63,86], lysine acetylation [87,88], lysine succinylation [88], or glycosylation [89] have been performed for intracellular media revealing PTMs on proteins involved in virulence.

Studies of PTMs on extracellular proteins are a research topic of interest in virulence but are less frequent than in intracellular media. Recently, Ser/Thr/Tyr phosphorylome [63], and both lysine acetylation and succinylation [23], have been investigated in the exoproteome of *P. aeruginosa* strain PA14. Taken together, these studies revealed that many virulence factors are modified (LasB, LasA, CbpD, azurin, or PrpL). The presence of multiple PTMs on a same effector protein, for example LasB or CbpD, two major virulence factors in this bacterium, were also detected. Interestingly, there were above 20 phosphorylation sites identified on LasB or CbpD in the extracellular medium while no phosphorylation was detected in the intracellular medium [23,63]. Similarly, the acetylated and succinylated lysine residues pointed out between the intra- and the extracellular media were not always the same [23,88]. Furthermore, for LasB, some lysine amino acids were detected modified with at least seven chemical groups (acetylation, succinylation, methylation, dimethylation, trimethylation, crotonylation, butyrylation, malonylation, and propionylation), meaning that a variety of proteoforms of LasB (and other effectors) exist. We can suggest that PTMs can be important for the secretion of proteins from the intracellular to the extracellular medium. Moreover, the diversity of chemical groups that can modify a residue can orient the protein function. LasB has an intracellularly function and multiple extracellular functions (tissue degradation, immune system avoidance, biofilm, etc.). The challenge is now to be able to detect all real proteoforms of these virulence factors by studying intact proteins [90,91].

The kinase NDK, secreted by either T1SS or T3SS, was also observed with different PTM patterns in the intracellular and the extracellular media: four succinylations and one acetylation in the intracellular milieu and only one succinylation in the extracellular milieu [23,88].

FliC constitutes the filaments of bacterial flagella of *P. aeruginosa*. The monomeric subunit FliC can also be translocated directly into the cytosol of the host cell via the T3SS [21,92]. Flagellins seem to be modified by a diversity of PTMs. It was detected, in the PAO1 strain, phosphorylated on Thr, Ser, and Tyr residues [93,94]. When no phosphorylation is present, the level of LasB, a T2SS-secreted protein, is increased. Flagellins in PAO1 were also reported glycosylated on Ser and Thr amino acids [95,96,97]. The role of these PTMs have to be determined with supplementary experiments.

Screening modified proteins in both intracellular and extracellular compartments is, thus, a complementary task since some of the proteins can be involved in virulence mechanisms, such as secretion system components, secretion system activation, toxins, or secreted effectors. In *P. aeruginosa*, it has been shown that phosphorylation is a key event in the T6SS activation. The kinase PpkA phosphorylates intracellularly the protein Fha1, triggering T6SS secretion [98,99]. The phosphatase PppA removes this phosphorylation. As shown above, intracellular proteins (GroEL, DnaK, catalase KatA) can be detected in the extracellular medium. These moonlighting proteins were also found with different PTMs, suggesting that PTMs can modulate their activities. For example, Hcp1, which constitutes the T6SS needle, was detected succinylated on lysine in intracellular medium [88]. Of note that enzymes involved in PTM addition or removal can also be of interest to study virulence in bacteria.

Virulence factors can also add PTMs on host proteins. In *P. aeruginosa*, ExoS and ExoT (T3SS) can modify different host target proteins to change their biological functions [100]. The characterisation of these mechanisms is challenging, but it is necessary to fully understand the virulence, to better understand the virulence of bacteria.

These results give many pieces of information on the modification states of secreted proteins by *P. aeruginosa*. Some examples have linked PTMs to virulence but the biological significance of many of these discovered PTMs on extracellular proteins have to be defined. These high-throughput identifications are a starting point for further studies (mutants, bacterial competition, native proteomics, etc.) to discover their roles in the virulence of *P. aeruginosa*. More than 250 modifications have been reported [101] and we can assume that new modifications that affect the role of proteins in virulence will be discovered in the next few years.

## 5. Proteomics for Host-Pathogen Studies

Proteomic analysis is an important resource for studying the response of the host and *P. aeruginosa* to infection. Several studies have focused on infection-associated antigens that trigger host immune responses. This approach is named immunoproteomics and can be achieved by combining 2-DE protein separation, Western blotting, and MS [102]. Immunoblotting was performed on *P. aeruginosa* strains isolated from cystic fibrosis (CF) patients [41]. Pooled human sera, in which the bacterium was not found, have been used for the detection of *P. aeruginosa* antigens, but without success. However, pooled sera from six CF-infected patients recognised several protein spots, but these sera revealed different proteins. The different strains may not produce any identified immunogenic proteins in secretomes in vivo. In 2008, Upritchard et al. focused on the proteins secreted by the strain PAO1 at the early stationary-phase [103]. After 2-DE migration and Western blotting, a total of 51 proteins were detected using sera from chronically infected patients. Four major exoproteins were identified: azurin, LasB, PrpL, and PasP. The authors also studied the strain Pa4 by immunoproteomics. The results revealed that there are some differences in the responses of different patient sera to *P. aeruginosa*. Immunoproteomics can therefore be a powerful first step for helping the identification vaccine antigens but it is necessary to understand host response to infection in order to optimise the protective response [104].

Studies using host models combined with proteomic approaches allow the analysis of the overall cellular processes and thus the understanding of the cellular and molecular changes occurring in the host and the pathogen during an infection. The researches described in the previous section mainly focused on laboratory strains and standard laboratory growth conditions that do not really mimic in vivo milieu. The goal is therefore to culture *P. aeruginosa* isolates in presence of host cells or in media close to in vivo conditions. The main problems encountered for these host-pathogen studies are both the recovery of sufficient amount of bacterial exoproteins from host milieu and the higher concentration of host proteins (e.g., albumin, mucin proteins). Various sample preparations have been developed [51] to improve these experiments. Several works have thus been published focusing either on the host side, the bacteria side or together. Host-pathogen relationships were studied in different models and some examples are described here.

Bastaert et al. investigated the activity of the protease LasB in alveolar macrophages cells [105]. They compared the exoproteome collected from these cells infected by *P. aeruginosa* PAO1 WT and Δ*lasB*. The data revealed that complement C3 and factor B proteins levels were lower in alveolar macrophages infected with *P. aeruginosa* PAO1 WT than in the mutant, suggesting a proteolytic effect of LasB on these components. Other experiments (as ELISA) were used to confirm and understand the role of LasB in the inactivation of innate immune system secreted components.

The nematode *Caenorhabditis elegans* can be used for host/*P. aeruginosa* interactions studies [106]. Recently, a DIGE approach combined with MALDI/MS was used to characterise proteins expressed by *C. elegans* during PAO1 infection at different times. It was shown that *P. aeruginosa* is able to inhibit protein translational events since the protein Eukaryotic elongation factor-2 (EEF-2) involved in protein elongation process was downregulated at 12 h and 24 h. The virulence factor exotoxin A could be responsible by binding to EEF-2. The proteins identified here revealed the different biological pathways used by *C. elegans* to fight the bacterial infection.

To identify markers of the host immune response and *P. aeruginosa* virulent factors important for the infection, Díaz-Pascual et al. chose a more complex living organism, zebrafish larvae as host model [107]. They compared the in vivo zebrafish larvae/*P. aeruginosa* strain PAO1 interactions in two types of contact: by injection of PAO1 in the zebrafish larvae or by immersion of the zebrafish larvae in a culture medium containing PAO1. The proteomic data highlighted differences in protein expression between the two types of infection: proteins associated with T3SS, flagellum, and drug response were more abundant when PAO1 was injected while proteins involved in single-species biofilm formation, pathogenesis, cellular response to antibiotic and starvation were over-expressed when PAO1 was in the environment. Moreover, host inflammatory response was only observed in the case of injection and epithelial or tissue response to PAO1 exoproteins was noticed in the case of immersion. The two kinds of contact give different pieces of information: injection seems to be adapted to study host response to infection whereas immersion is more appropriate to study bacterial virulence. The zebrafish embryo is a suitable first model for studying *P. aeruginosa* infections [108] for next investigating murine models.

To examine the signalling pathway activation in mice after different Toll-like receptors (TLRs) stimulations, Koppenol-Raab et al. performed a quantitative analysis with stable isotope labelling by amino acids in cell culture (SILAC) of extracellular proteins of TLR-stimulated murine macrophages after or without PAO1 infection [109]. TLRs are predominantly expressed by innate immune cells such as macrophages and dendritic cells. They enable the recognition of diverse microbial molecules and trigger the production of proteins involved in the host immune response. These proteomic results were compared to transcriptomic analyses. The data showed a strong correlation between the transcriptomic and exoproteomic data for TLR2 and TLR7 stimulations.

*P. aeruginosa* is involved in eye infection. With a quantitative approach, Yeung et al. investigated for the first time the exoproteome of eye wash in mice after biofilm formation of the invasive *P. aeruginosa* clinical isolate 6294 [110]. They succeeded to detect bacterial proteins at the corneal surface that are involved in both bacterial virulence and survival. They also accessed to host responses by comparing two conditions: control samples with infected samples to gain insight on host immune response to infection, and infected samples with counter lateral samples for information on the direct response to infection at the host-pathogen interface site.

*Staphylococcus aureus* and *P. aeruginosa* are commonly found together, causing serious acute and chronic infections, including pneumonia, urinary tract infections, and chronic wounds [111,112]. In CF patients, during the adolescence and early adulthood, *P. aeruginosa* replaces *S. aureus* but sometimes this latter is still detected in CF adults. Pallet et al. studied the co-infection of CF airways by *S. aureus* and *P. aeruginosa* under normoxia and anoxia [113]. Amongst three CF isolates, one presented no anti-staphylococcal activity. To identify virulence factors involved in the dominance of *P. aeruginosa*, they compared the exoproteomes of the laboratory strain PAO1 with the three CF isolates. Nevertheless, no candidate was highlighted, and the authors suggested that other molecules than proteins could be involved in the anti-staphylococcal activity.

Biofilms are an important bacterial state increasing bacterial antibiotic resistance. Extracellular proteins can play an important role in biofilm development. With approaches other than proteomics, Tielen et al. have previously shown the role of extracellular enzymes (in particular LipA, LipC, EstA, and LasB) on differentiation of *P. aeruginosa* SG81 strain biofilms [114]. However, biofilms are usually formed by multiple microorganisms competing for nutrients and survival. The exoproteomes of *P. aeruginosa* biofilm, the fungus *C. albicans* biofilm and *P. aeruginosa* and *C. albicans* mixed biofilm were analysed by MALDI-MS/MS at different times [115]. In this study, 16 proteins (including exotoxin A, pyoverdine, and proteins involved in iron acquisition systems) among identified virulence factors were more abundant or only observed in the mixed biofilm. These proteomic results indicate that in biofilm *P. aeruginosa* competes with *C. albicans* for nutrients.

## 6. Conclusions

*P. aeruginosa* is involved in a wide range of human diseases, including eye infections, CF lung infections or nosocomial infections. This pathogen must adapt specifically to the host environment and uses secreted virulence factors to degrade host tissues or to neutralise the host immune response. The characterisation of exoproteins is therefore a hot topic in order to understand the mechanisms set up by the bacterium and to determine new therapeutic targets for the development of drugs or antigens for vaccine design.

Proteomics is now a well-established approach to gain a deeper understanding of the molecular mechanisms involved in the pathogenicity of *P. aeruginosa* by identifying and quantifying exoproteins in different conditions. This proteomic research clearly shows that this bacterium disseminates an arsenal of virulence factors to promote infection. A microbial genome study predicted that 25–35% of proteins could be exported [32]. Furthermore, a *P. aeruginosa* strain with deleted known effector genes was observed still virulent during acute lung infection [116]. Thus, several virulence factors are still unknown, and their identifications require further experiments. Exoproteomics is therefore an important tool to identify them. The last exoproteomes studies have thus pointed out new potential secreted proteins (like hypothetical proteins) that seem to be involved in *P. aeruginosa* virulence. Further experiments (molecular biology tools, microbiological assays, microscopy, immunological techniques, etc.) will be used to reveal in which precise mechanisms they play a role. These exoproteomic studies also highlight the complex adaptative and survival strategies used by *P. aeruginosa* to modulate immunity at the site of infection.

PTM investigations is also an interesting research topic. Different virulence factors were described modified with phosphorylation, acetylation, succinylation and other chemical groups. Future improvements in MS will be an advantage in increasing the number of both identification and PTMs of virulence factors. Studying PTMs in the context of the host-pathogen relationship, while challenging, would allow us to see their influence on virulence and infection. Moreover, investigations on biofilm exoproteomes (without and with other bacteria to mimic microbiota for example) have to be conducted to understand the bacterial competition and prevalence of some bacteria during chronic infections.

The study of the intracellular compartment should not be overlooked to identify virulence factors. Virulence factors are taken by chaperones for their export and investigating chaperones partners can be informative for the discovery of new effectors [16,39]. Components of T3SS and T6SS export apparatus were also observed with different PTMs [50]. Without *N*-glycosylation, the virulence factor LecB cannot be exported in *P. aeruginosa* periplasm [117]. Intracellular investigations can thus be a source of informative data for inhibiting bacterial virulence.

Understanding host-pathogen interactions is crucial for the development and improvement of treatments against pathogenic microorganisms and the discovery of new vaccine candidates [118,119]. Proteomics coupled with other omic approaches (genomics, transcriptomics, interactomics, immunoproteomics, or metaproteomics) extends the repertoire of tools to study pathogen-induced infections [120]. An integrative omic approach will be a solution to find new bacterial mechanisms and inhibitory pathways to combat *P. aeruginosa* virulence.

## Figures and Tables

**Figure 1 toxins-12-00571-f001:**
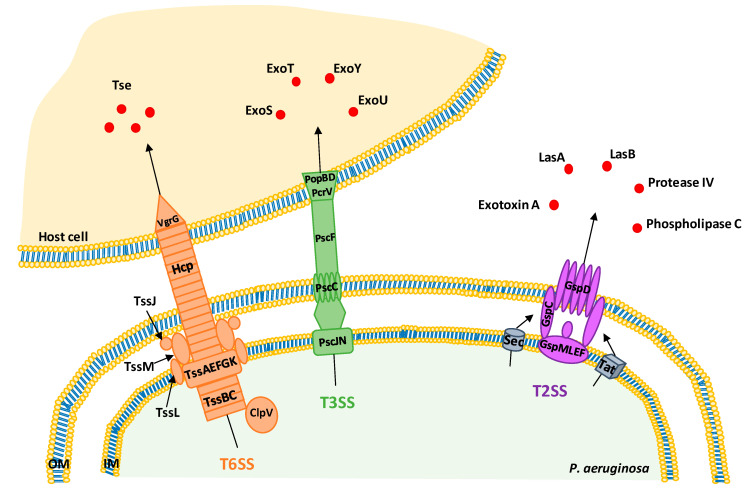
Schematic representation of type 2, 3, and 6 secretion systems (T2SS, T3SS, and T6SS) in *P. aeruginosa*. OM: outer membrane; IM: inner membrane.

**Figure 2 toxins-12-00571-f002:**
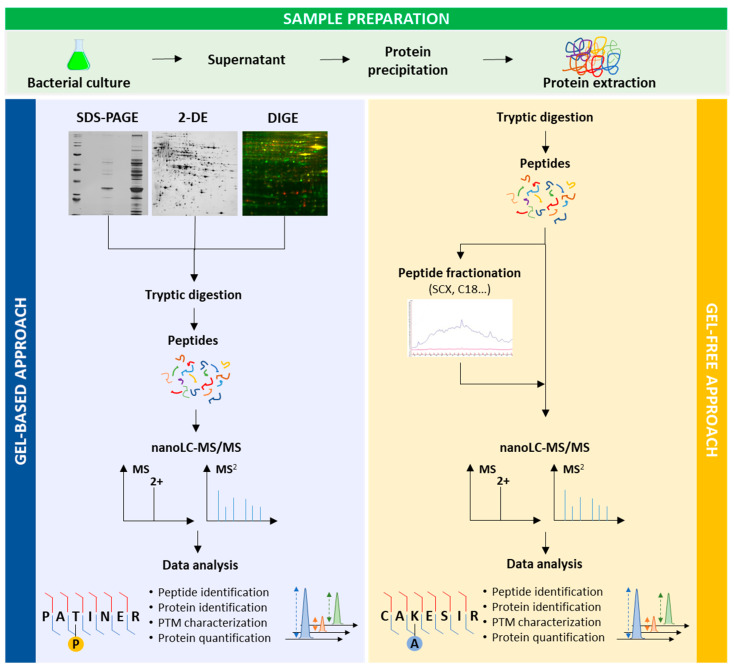
Strategies for gel-based and gel-free proteomic approaches for the study of exoproteins.

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
