# Peer review of "Exoproteomics for Better Understanding Pseudomonas aeruginosa Virulence"

_toxins, 2020, doi:10.3390/toxins12090571_

Round 1
Reviewer 1 Report
In this review, the authors describe research into the exoproteome of Pseudomonas aerigunosa, a bacterium associated with various nosocomial diseases. This area is a lesser studied niche within the broader field of proteomics, which appears mostly due to the technical challenge of protein isolation and enrichment from culture supernatant and infected host tissues. The central thesis is that characterization of the exoproteome may lead to novel antimicrobial strategies, the identification of biomarkers of infection, and a broader understanding of host-microbe interactions. These areas are buried in various parts of the paper, and instead the authors mostly refer to using exoproteomics to “fight” against this pathogen, which is a vague and weak placeholder for what the studies described and exoproteomics in general add to the understanding of the pathosystem. I think the manuscript would be vastly strengthened by a paragraph that explains up front why this area should be explored, what it can contribute to the field, and why it is largely overlooked. Additionally, the article needs to be edited for English language and grammar, and it is suggested that the authors work with a native English speaker to revise prior to resubmission. Below are some of the language problems and typos I’ve found, but it is not a comprehensive list, as well as a few minor comments/clarifications that are requested from the authors: Minor edits: Line 13: host-pathogen relationships (plural, not singular) Line 19: “non-sporing, and ubiquitous because of it’s ability to survive…” Lines 27-29: this sentence is confusing. How can you eliminate a pathogen without killing it? What does this have to do with reducing the selection pressure of antibiotics? I can kind of see what you are trying to say, but it is written in a convoluted manner Paragraph beginning at line 32: I suggest that the authors break this up into two paragraphs – one describing QS and another the secretion systems. QS could also be expanded upon, given the large amount of genomic content that is controlled by the system. Line 45 and Fig. 1: exotoxin is misspelled Line 54: unknown is misspelled Line 57: “providing fitness and survival advantages…” OR “providing a fitness and survival advantage…” Line 93-95: Another confusing sentence with improper grammar. Why do we need to consider technological developments? This is vague. Reading further, I see that the authors mean that researchers need to choose the correct sample prep, instrumentation/mass spectrometer, and data analysis parameters for their experimental goals. The consideration of “…useful information obtained on protein identification involved in molecular mechanisms” is similarly vague. Line 106: Again, bacterial mechanism overviews is vague. Mechanisms for what? I suggest instead wording it something along the lines of “…in different conditions, which have contributed broadly to the understanding of bacterial physiology and virulence. However, these studies miss an important subset of the proteome – the subset that are secreted into the extracellular milieu.” Line 116: analyze, not analyse Line 135, 140: “any prior knowledge” Lines 144-146: Why is calcium depleted media not also used for T6SS activation, which also needs host cell contact? Line 173: bottom is misspelled Line 297: “…Tse2 to compete with bacterial cells…” Line 302: “A higher secretion level…” OR “Higher secretion levels…” Line 305: “For the latter, …” Line 341: Revise sentence for grammar Line 349: “…as secreted, such as…” Line 358: Can the authors comment on how many of the hypotheticals are found intracellularly? Is the percentage lower than what is seen in the extracellular space? Line 413-414: How are the LasA or LasB proteins different when the flagellin is not phosphorylated? Do they have different PTMs? Different levels? Line 429: Revise sentence for grammar (“interesting research way” is poor wording and vague, suggest changing to “…but it is necessary to fully understand the virulence…”) Lines 470-471: Confusing and incomplete sentence, please revise Line 526: “…competes with C. albicans…” Line 537: “A microbial genome study predicted that…” Line 566: pathways is misspelled Table 1. A lot of these omics approaches are not mentioned at all in the paper, and defining them here seems to come out of nowhere. The authors should probably expand on the strategies in the introduction or in the final paragraph; otherwise, this table feels like it does not belong in the paper.Author Response
Please see the attachment.

Reviewer 2 Report
Thus, 27 eliminating pathogens without killing them can be a promising antimicrobial strategy that at the 28 same time reduces the selection pressure of antibiotics on bacteria.
The above line concerns me. If I am understanding the authors' suggestion correctly, they are arguing that targeting virulence factors can reduce the impact of a pathogen without the risk of creating more multi-drug-resistant or antibiotic resistant PA. While it may be true that ones is less likely to generate an antibiotic resistance bacterium, mitigating the virulence potential of a pathogen will make it less fit. Evolutionary principles apply across the board. It is still possible to select for isolates with virulence factors not impacted by a given treatment. Many antibiotics target specific pathways or proteins necessary for life. Targeting proteins necessary for virulence may be a different selective pressure, but it is still a selective pressure.
Additionally, the sentence structure in the abstract makes this difficult to read.
"P. aeruginosa possesses a large arsenal of virulence systems contributing to the success of its 32 infection and colonization [6]. Among them, we can cite quorum sensing (QS) and type 2, 3 and 6 33 secretion systems (T2SS, T3SS and T6SS respectively) with their secreted proteins and toxins. QS is 34 an intercellular bacterial communication system dependant of cell density which allows individual 35 cells to act as a community [4]."
Dependent on, not of, unless you meant to say "independent" which would make the statement false.
It is clear that the author(s) are very knowledgeable about PA virulence. In that regard, it is a joy to read. I am learning, or relearning things.
Cofilin is mentioned, off the cuff, as though the authors expect everyone knows what that is without preamble. Give us some preamble.
This review does an excellent job at showcasing the power and potential of MS in the study of bacterial proteomes. It covers the bulk of the proteomic approaches extant and relates it well to a relevant public health concern.
There are significant grammar and sentence structure issues that need to be addressed, but aside from that the content is excellent.
I feel the authors missed sighting some significant work, such as the paper by Fagerquist et al. 2018 in Frontiers, which show cased the first ever report of trypsin inducing the virulence response in Salmonella as well as differential regulation between serovars. Such findings can also inform protease selection when working with live cells for MS purposes.
Round 2
Reviewer 1 Report
The paper reads much better and I feel the major concerns have been sufficiently addressed. It would still benefit from being proofread by a native English speaker; however, I understand this is not always possible. The paper as it stands is in an acceptable format for publication.